# SARS-CoV-2 rapid antibody test results and subsequent risk of hospitalisation and death in 361,801 people

Matthew Whitaker[1,2,10], Bethan Davies [1,2,3,10], Christina Atchison [1,3,4], Wendy Barclay [4,5], Deborah Ashby [1], Ara Darzi [3,4], Steven Riley[6], Graham Cooke [3,4,5], Christl A. Donnelly [1,6,7], Marc Chadeau-Hyam [1,2], Paul Elliott [1,2,3,4,8,9] ✉ & Helen Ward [1,3,4,6]

The value of SARS-CoV-2 lateral flow immunoassay (LFIA) tests for estimating individual disease risk is unclear. The REACT-2 study in England, UK, obtained self-administered SARS-CoV-2 LFIA test results from 361,801 adults in January-May 2021. Here, we link to routine data on subsequent hospitalisation (to September 2021), and death (to December 2021). Among those who had received one or more vaccines, a negative LFIA is associated with increased risk of hospitalisation with COVID-19 (HR: 2.73 [95% confidence interval: 1.15,6.48]), death (all-cause) (HR: 1.59, 95% CI:1.07, 2.37), and death with COVID-19 as underlying cause (20.6 [1.83,232]). For people designated at high risk from COVID-19, who had received one or more vaccines, there is an additional risk of all-cause mortality of 1.9 per 1000 for those testing antibody negative compared to positive. However, the LFIA does not provide substantial predictive information over and above that which is available from detailed socio-demographic and health-related variables. Nonetheless, this simple test provides a marker which could be a valuable addition to understanding population and individual-level risk.

Lateral flow immunoassay tests (LFIAs) have been used at a population level to track antibody prevalence[1] and quantify waning antibody levels post vaccination[2]. The clinical value of these simple rapid tests at an individual level is less clear, although they have reasonable sensitivity and specificity in detecting SARS-CoV-2 antibodies at a threshold relevant to viral neutralisation[3]. Clinical use of LFIAs has been suggested for tracking seroconversion post-infection[4] and post-vaccination[5], and monitoring waning antibody levels in the months following vaccination[6], for example in those with immunosuppression[7]. Further, IgG antibody positivity using self-administered LFIAs has been shown to

predict COVID-19 infection risk over a 200-day follow-up period in an unvaccinated population[8]. Nonetheless, to date there has been limited uptake of antibody LFIAs at an individual level to guide understanding of personal risk from COVID-19 and inform decision-making about behaviour and need for boosters or other protections.

The REal-time Assessment of Community Transmission-2 (REACT-2) study was a nationwide study of SARS-CoV-2 antibody prevalence in adults in England in which randomly selected adults were invited to complete an LFIA test at home and to complete a short online or telephone survey[9]. Here we follow up data from adults who took part in

[1]School of Public Health, Imperial College London, London, UK. [2]MRC Centre for Environment and Health, Imperial College London, London, UK. [3]Imperial College Healthcare NHS Trust, London, UK. [4]National Institute for Health Research Imperial Biomedical Research Centre, London, UK. [5]Department of Infectious Disease, Imperial College London, London, UK. [6]MRC Centre for Global Infectious Disease Analysis and Jameel Institute, Imperial College London, London, UK. [7]Department of Statistics, University of Oxford, Oxford, UK. [8]Health Data Research (HDR) UK London at Imperial College, London, UK. [9]UK Dementia Research Institute at Imperial College, London, UK. [10]These authors contributed equally: Matthew Whitaker, Bethan Davies. ✉e-mail: p.elliott@imperial.ac.uk

the final two rounds of REACT-2, conducted in January–May 2021, after the start of the national vaccination campaign, using their linked National Health Service (NHS) administrative health records. We use these data to assess whether LFIA tests have predictive value for adverse health outcomes in the general population, and whether they might be useful in a clinical setting. Specifically, we explore whether LFIA results are associated with future hospitalisation (all-cause, emergency and COVID-19) and mortality (all-cause and COVID-19).

## Results

The study population included 361,801 participants, of whom 143,774 (39.7%) tested positive on self-administered LFIAs in rounds 5 and 6 (January–February 2021 and May 2021, respectively) of the REACT-2 study. Table 1 shows the characteristics of the study population overall and by "high" or "low" COVID-19 risk based on the UK Joint Committee on Vaccination and Immunisation (JCVI) definition of being clinically extremely vulnerable for COVID-19 (see Supplementary Methods, S.1). The mean age was 55.5 years; 31.4% were aged 50 years or under; and 56.0% were women. A single vaccination was reported by 28.9% of participants and two vaccinations by 24.3%. Over 1 in 6 (15.8%) reported prior COVID-19, although only 24.9% of these cases had been confirmed (e.g. by polymerase chain reaction (PCR) test) as most occurred in the first COVID-19 wave when community testing was limited.

Among the 192,637 people who had received one or more vaccinations at least 14 days prior to the LFIA test, 128,282 (66.6%) were LFIA positive; 88,026 people had received two vaccinations, of whom 74,534 (84.7%) tested positive. Of the 14,271 with prior COVID-19 confirmed with a test, 12,065 (84.5%) were LFIA positive; of the 57,317 who reported confirmed or suspected prior COVID-19, 29,700 (51.8%) were LFIA positive.

We obtained follow-up data on hospitalisations to September 30, 2021, with mean follow-up time of 184 days and a total of 182,659 person years of follow-up. For mortality we obtained data to December 1, 2021, with a mean follow-up time of 245 days and a total of 243,125 person years of follow-up.

For hospitalisations, 16,802 (4.6%) people were admitted for any cause (excluding pregnancy and accident/injury related, see "Methods"), 5330 of whom were unplanned (emergency) admissions, 91 had COVID-19 as the primary diagnosis and 151 had COVID-19 listed anywhere on their hospital record. There were 389 deaths, including 19 with COVID-19 as the underlying condition and 21 with COVID-19 mentioned anywhere on the death certificate (Table 1).

In N = 25,824 high-risk individuals with one or more vaccinations, those testing LFIA negative had a risk of all-cause mortality of 3.6, compared to 1.7 per 1000 in those testing LFIA positive, an absolute difference of 1.9 per 1000 (Supplementary Table 1).

Figure 1 shows Kaplan–Meier plots of the cumulative hazard functions for hospitalisations and deaths in individuals who had received one or more vaccinations (N = 192,604). Those testing negative on the LFIA antibody test had higher cumulative hazards for all outcomes at almost all time points during follow-up. Findings were similar when stratified by risk group, by age and by vaccination status (Fig. 1 and Supplementary Fig. 1).

In mutually adjusted Cox regression models accounting for age, sex, risk group, prior COVID-19 and vaccination count, those with a negative LFIA after one or more vaccinations had an increased mortality from all causes (HR: 1.59 [1.07, 2.37], and mortality with COVID-19 as underlying cause (HR: 20.6 [1.83, 232]), as well as increased risk of hospitalisation with COVID-19 as the primary diagnosis (2.73 [1.15, 6.48]).

After two or more vaccinations, those with a negative LFIA had increased all-cause mortality (HR: 1.87 [1.11, 3.13]), mortality with COVID-19 as underlying cause (HR: 12.46 [1.13, 138]), hospitalisation with COVID-19 as the primary diagnosis (2.79 [1.01, 7.72]) and

hospitalisation with COVID-19 mentioned (2.54 [1.01, 6.41]) compared to people who tested positive (Table 2 and Supplementary Tables 2 and 3).

### Stability selection

We used 100x subsampled Least Absolute Shrinkage and Selection Operator (LASSO)-penalised Cox models with demographic, biological and health variables together with LFIA result to identify which were preferentially selected as predictive of outcomes (see "Methods"). In these multivariable models, LFIA result was not selected as a predictor of subsequent hospitalisation, emergency hospitalisation, hospitalisation with COVID-19 or death in the population with one or more vaccines, nor when stratified by risk group (Fig. 2). The main positive predictors selected were risk group, age (continuous), anaemia, smoking and body mass index (BMI, continuous).

### Boosted tree models

We used boosted tree models trained on different subsets of predictors, with and without LFIA, cross-validated on unseen data, to assess the predictive value of LFIA result over and above the other variables (age, sex, risk group, prior COVID-19 history and vaccination count, see "Methods") in the population who had received one or more vaccines. Adding LFIA result to the boosted tree model did not substantially improve the prediction of hospitalisation or death (Fig. 3); however predictive accuracy was improved by including additional survey data (BMI, smoking status, deprivation, education status, comorbidities) to the model. The findings were consistent when analysis was repeated in the full population (Supplementary Fig. 3).

## Discussion

In this large, representative study of the population of England, we found that a negative LFIA antibody test result from early 2021, after receiving one or more vaccines at least 14 days previously, was associated with increased risk of hospitalisation from COVID-19 and of death from all causes and those attributed to COVID-19 over the subsequent 6–9 months. These associations persisted after adjustment for multiple factors including age. However, a combination of variable selection approaches and tree-based predictive modelling indicated that the information contained in the LFIA result could be adequately captured by the responses to a set of survey questions on basic demographics and health status.

Nonetheless, the LFIA test might provide a simple and rapid means to identify those at increased risk who might benefit from additional booster doses or rapid access to antiviral medication. The LFIA is a threshold test which correlates with SARS-CoV-2 antibody titres and live virus neutralisation[10] and negativity following vaccination or infection may be a biomarker of a limited specific immune response[11] and of frailty[12] and therefore may be linked to hospitalisation and all-cause mortality. People with immunosuppression have lower antibody titres post-vaccination, and approximately one in five people with solid organ transplants, rare autoimmune rheumatic diseases and lymphoid malignancies are LFIA negative after three vaccine doses[13].

In addition, the LFIA provides information, which can be obtained rapidly and at scale, on population antibody levels which may inform the timing and prioritisation of vaccination booster campaigns[14].

Our study has limitations. LFIAs have imperfect accuracy: the tests used in this study were reported to have sensitivity of 78.7% (95% confidence interval [CI]: 71.8, 84.6)[10]; therefore, our LFIA test results will likely include some false negatives which may have reduced (biased) our risk estimates for hospitalisations and mortality following a negative test post-vaccination. Furthermore, the sensitivity of LFIAs may be lower in the weeks following vaccination; although we considered here a vaccine dose to be effective 14 days after injection, we cannot rule out the possibility of bias from time-since-vaccination,

**Table 1 | Population characteristics, by risk group status and LFIA result**

| Variable | Category | High-risk group | | Low-risk group | | Full cohort |
|---|---|---|---|---|---|---|
| | | LFIA− | LFIA+ | LFIA− | LFIA+ | |
| All participants | All participants | 28,368 (100%) | 26,810 (100%) | 189,659 (100%) | 116,964 (100%) | 361,801 |
| Age | Mean (SD) | 60.86 (13.19) | 64.29 (10.84) | 51.04 (15.24) | 59.4 (13.25) | 55.49 (14.97) |
| Age group | 50 and under | 5229 (18.4%) | 2353 (8.8%) | 82,382 (43.4%) | 23,299 (19.9%) | 113,263 |
| | 51–70 | 16,381 (57.7%) | 16,468 (61.4%) | 90,628 (47.8%) | 71,290 (61%) | 194,767 |
| | 71 and over | 6758 (23.8%) | 7989 (29.8%) | 16,649 (8.8%) | 22,375 (19.1%) | 53,771 |
| Sex | Female | 13,994 (49.3%) | 13,858 (51.7%) | 103,652 (54.7%) | 71,181 (60.9%) | 202,685 |
| | Male | 14,373 (50.7%) | 12,951 (48.3%) | 86,003 (45.3%) | 45,781 (39.1%) | 159,108 |
| Ethnicity | Asian | 571 (2%) | 837 (3.1%) | 5180 (2.8%) | 3980 (3.4%) | 10,568 |
| | Black | 239 (0.8%) | 331 (1.2%) | 1402 (0.7%) | 1326 (1.1%) | 3298 |
| | Mixed | 208 (0.7%) | 202 (0.8%) | 2221 (1.2%) | 993 (0.9%) | 3624 |
| | Other | 194 (0.7%) | 264 (1%) | 1364 (0.7%) | 1094 (0.9%) | 2916 |
| | White | 26,914 (95.7%) | 24,970 (93.9%) | 178,117 (94.6%) | 108,753 (93.6%) | 338,754 |
| Vaccination status | 0 vaccines | 17,828 (62.8%) | 1526 (5.7%) | 135,844 (71.6%) | 13,966 (11.9%) | 169,164 |
| | 1 vaccine | 6197 (21.8%) | 7032 (26.2%) | 44,665 (23.6%) | 46,716 (39.9%) | 104,610 |
| | 2 vaccines | 4343 (15.3%) | 18,252 (68.1%) | 9150 (4.8%) | 56,282 (48.1%) | 88,027 |
| Body mass index | Mean (SD) | 27.93 (6.11) | 27.81 (5.92) | 26.76 (5.26) | 26.75 (5.17) | 26.93 (5.37) |
| Index of multiple deprivation (IMD) quintile | 1—most deprived | 3432 (12.1%) | 2983 (11.1%) | 17,055 (9%) | 9623 (8.2%) | 33,093 |
| | 2 | 4988 (17.6%) | 4481 (16.7%) | 29,316 (15.5%) | 17,468 (14.9%) | 56,253 |
| | 3 | 6204 (21.9%) | 5747 (21.4%) | 41,022 (21.6%) | 24,786 (21.2%) | 77,759 |
| | 4 | 6729 (23.7%) | 6653 (24.8%) | 47,786 (25.2%) | 30,233 (25.8%) | 91,401 |
| | 5—least deprived | 7015 (24.7%) | 6946 (25.9%) | 54,480 (28.7%) | 34,854 (29.8%) | 103,295 |
| Smoking status | Current cigarette smoker | 3344 (11.8%) | 2104 (7.8%) | 17,815 (9.4%) | 6486 (5.5%) | 29,749 |
| | Not current cigarette smoker | 24,739 (87.2%) | 24,413 (91.1%) | 169,980 (89.6%) | 109,442 (93.6%) | 328,574 |
| | Prefer not to say | 285 (1%) | 293 (1.1%) | 1864 (1%) | 1036 (0.9%) | 3478 |
| Previous case of COVID | No known previous COVID-19 | 25,036 (88.3%) | 22,385 (83.5%) | 165,374 (87.2%) | 91,689 (78.4%) | 304,484 |
| | Yes, confirmed by doctor but not tested | 340 (1.2%) | 416 (1.6%) | 1260 (0.7%) | 1337 (1.1%) | 3353 |
| | Yes, confirmed by positive test | 243 (0.9%) | 1710 (6.4%) | 1963 (1%) | 10,355 (8.9%) | 14,271 |
| | Yes, my own suspicions | 2749 (9.7%) | 2299 (8.6%) | 21,062 (11.1%) | 13,583 (11.6%) | 39,693 |
| Organ transplant recipient | Yes | 320 (1.1%) | 161 (0.6%) | 0 (0%) | 0 (0%) | 481 |
| Diabetes (type I, type II or gestational) | Yes | 4028 (14.2%) | 4067 (15.2%) | 7874 (4.2%) | 6928 (5.9%) | 22,897 |
| Heart disease or heart problems | Yes | 8699 (30.7%) | 8804 (32.8%) | 0 (0%) | 0 (0%) | 17,503 |
| Hypertension (high blood pressure) | Yes | 7295 (25.7%) | 7657 (28.6%) | 21,262 (11.2%) | 19,814 (16.9%) | 56,028 |
| Stroke | Yes | 1386 (4.9%) | 1126 (4.2%) | 0 (0%) | 0 (0%) | 2512 |
| Kidney disease | Yes | 1592 (5.6%) | 1499 (5.6%) | 0 (0%) | 0 (0%) | 3091 |
| Liver disease | Yes | 917 (3.2%) | 780 (2.9%) | 0 (0%) | 0 (0%) | 1697 |
| Anaemia | Yes | 900 (3.2%) | 734 (2.7%) | 2062 (1.1%) | 1031 (0.9%) | 4727 |
| Asthma | Yes | 4120 (14.5%) | 3803 (14.2%) | 16,521 (8.7%) | 9955 (8.5%) | 34,399 |
| Other lung condition (e.g. COPD, bronchitis or emphysema) | Yes | 4962 (17.5%) | 5062 (18.9%) | 0 (0%) | 0 (0%) | 10,024 |
| Cancer | Yes | 3482 (12.3%) | 3385 (12.6%) | 0 (0%) | 0 (0%) | 6867 |
| Condition affecting the brain and nerves | Yes | 2294 (8.1%) | 1728 (6.4%) | 0 (0%) | 0 (0%) | 4022 |
| A weakened immune system | Yes | 7886 (27.8%) | 6371 (23.8%) | 0 (0%) | 0 (0%) | 14,257 |
| Depression | Yes | 3430 (12.1%) | 2505 (9.3%) | 13,098 (6.9%) | 5824 (5%) | 24,857 |
| Anxiety | Yes | 3772 (13.3%) | 2875 (10.7%) | 20,034 (10.6%) | 9059 (7.7%) | 35,740 |
| Psychiatric disorder | Yes | 351 (1.2%) | 234 (0.9%) | 1091 (0.6%) | 410 (0.4%) | 2086 |
| Shielding out of concern over COVID-19 | Yes | 12,808 (45.1%) | 8423 (31.4%) | 16,722 (8.8%) | 6195 (5.3%) | 44,148 |
| Declared clinically vulnerable by medical professional | Yes | 9740 (34.3%) | 10,359 (38.6%) | 0 (0%) | 0 (0%) | 20,099 |
| **Outcomes** | | | | | | |
| Hospital admission (any cause) | Yes | 3803 (13.4%) | 1677 (6.3%) | 8192 (4.3%) | 3130 (2.7%) | 16,802 |
| Hospital admission (emergency) | Yes | 1335 (4.7%) | 524 (2%) | 2571 (1.4%) | 900 (0.8%) | 5330 |

**Table 1 (continued) | Population characteristics, by risk group status and LFIA result**

| Variable | Category | High-risk group | | Low-risk group | | Full cohort |
|---|---|---|---|---|---|---|
| | | LFIA− | LFIA+ | LFIA− | LFIA+ | |
| Hospital admission due to COVID-19 (primary diagnosis) | Yes | 20 (0.1%) | 9 (0%) | 51 (0%) | 11 (0%) | 91 |
| Hospital admission with COVID-19 (any diagnosis) | Yes | 30 (0.1%) | 18 (0.1%) | 80 (0%) | 24 (0%) | 152 |
| Death (any cause) | Yes | 173 (0.6%) | 51 (0.2%) | 135 (0.1%) | 30 (0%) | 389 |
| Death with COVID-19 as underlying condition[a] | Yes | 10 (0%) | <8 (0%) | <8 (0%) | 0 (0%) | 20 |
| Death with COVID-19 mentioned on death certificate[a] | Yes | 10 (0%) | <8 (0%) | 10 (0%) | 0 (0%) | 20 |

High-risk group status is based on the UK JCVI definition of clinically extremely vulnerable using self-reported comorbidity information.
[a]Small number suppression and rounding applied.

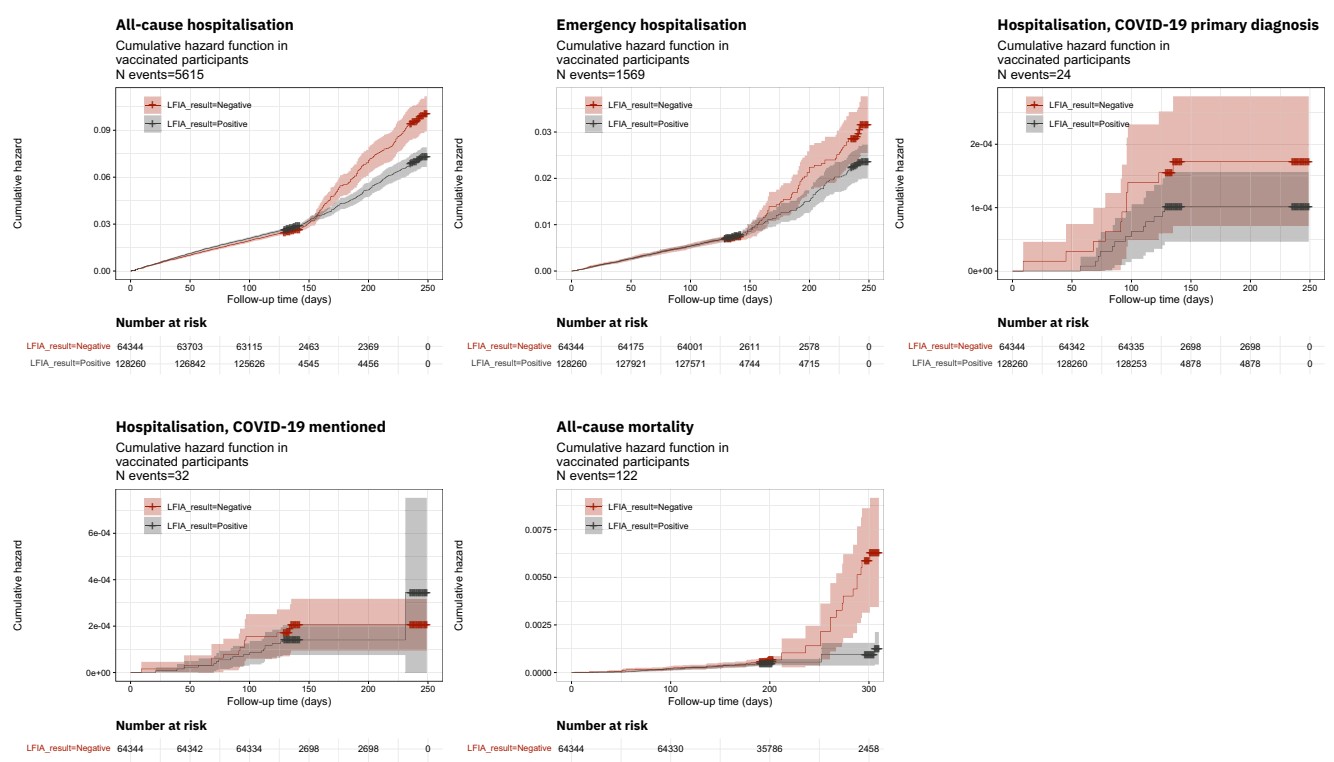

**Fig. 1 | Kaplan–Meier plots showing cumulative hazard among people reporting a positive test on the LFIA (grey line) and those reporting a negative test (red line) for (from top left to bottom right) (i) all-cause hospitalisation, (ii) emergency hospitalisation, (iii) hospitalisation with COVID-19 as the primary diagnosis, (iv) hospitalisation with COVID-19 mentioned anywhere on the hospital record, and (v) all-cause mortality, among N = 192,604 participants who had received one or more vaccines.** 95% pointwise confidence intervals are shown in shaded colour.

**Table 2 | Hazard ratios from multiple Cox regression models**

| | Whole population n = 361,801 | One or two vaccines n = 192,637 | Two vaccines n = 88,027 |
|---|---|---|---|
| All-cause death | 1.53 [1.10,2.13] | 1.59 [1.07,2.37] | 1.87 [1.11,3.13] |
| Death with COVID underlying cause | 12.88 [1.40,118.56] | 20.58 [1.83,231.89] | 12.46 [1.13,137.67] |
| Hospitalisation | 0.97 [0.93,1.02] | 1.04 [0.98,1.10] | 1.07 [0.97,1.18] |
| Emergency hospitalisation | 0.96 [0.89,1.05] | 1.05 [0.94,1.18] | 1.19 [1.00,1.43] |
| Hospitalisation with COVID-19[a] | 1.92 [0.99,3.71] | 2.73 [1.15,6.48] | 2.79 [1.01,7.72] |

Each outcome is modelled as the dependent variable, and adjustment on age, sex, prior COVID-19 infection (and vaccination status), among those with one or more vaccinations and those with two vaccinations, and the full cohort. The independent variable of interest is the LFIA test result. 95% confidence intervals are shown in square brackets.
[a]As primary diagnosis.

which is related to age and comorbidities[15]. In addition, self-administered LFIAs depend on user interpretation of the finding; nonetheless, validation analysis of the user-interpreted results from REACT-2 showed substantial concordance between user submitted results and those generated both by experts (Cohen's kappa: 0.89) and by a computer vision machine learning algorithm (Cohen's kappa: 0.80)[16–18]. There are also limitations to the routine data on outcomes including the duration of follow-up which was limited so far to between 6 and 9 months; in addition hospital and mortality data are from the monthly release of provisional data, while full quality assurance is only provided for annual extracts. It is therefore possible that some misclassification may have occurred. A further limitation is possible recruitment bias. Although the REACT-2 study included a large random population sample[2], there may have been differential participation linked to risk of severe outcomes.

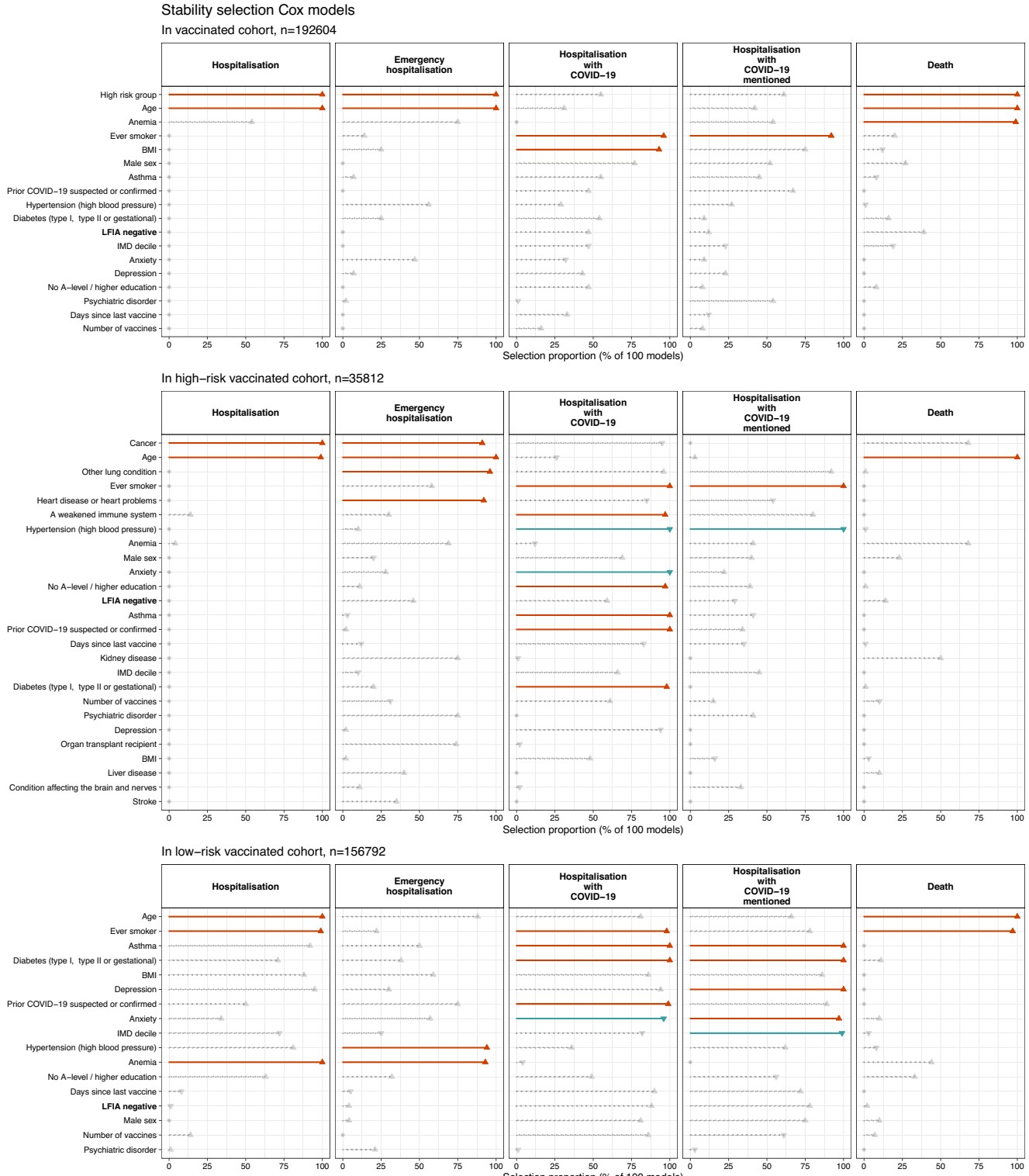

**Fig. 2 | Results of penalised Cox modelling with stability selection.** Results shown for five main outcomes (l-r: hospitalisation, emergency hospitalisation, hospitalisation with COVID-19, hospitalisation with COVID-19 mentioned, death) and for vaccinated (one or more) cohort (top, $N = 192,604$), high-risk vaccinated cohort (middle, $N = 35,812$) and low-risk vaccinated cohort (bottom, $N = 156,792$).

In conclusion, a negative LFIA result following vaccination is a marker of elevated risk of hospitalisation and death over a 6–9-month time horizon. For people designated at high risk from COVID-19, there was an additional risk of all-cause mortality of 1.9 per 1000 for those testing antibody negative compared to positive. However, the LFIA does not provide substantial predictive information over and above that which is available from detailed sociodemographic and health-related variables. Nonetheless, this simple test provides a marker which could be a valuable addition to understanding population and individual level risk.

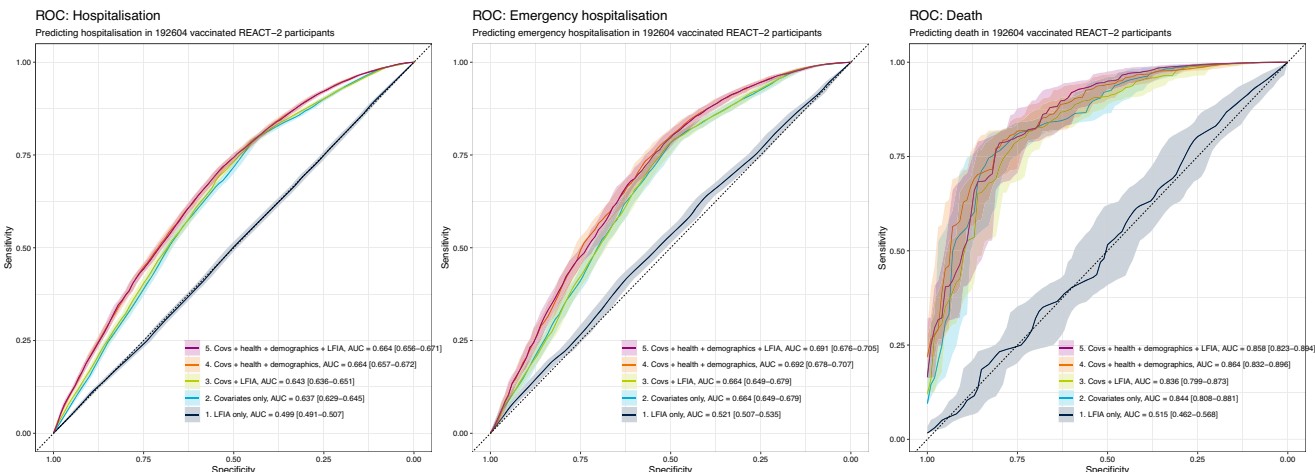

**Fig. 3 | ROC curves quantifying the additional predictive power conferred by adding LFIA test result to other survey data from REACT-2 (age, sex, BMI, smoking status, education, deprivation, vaccination status, prior COVID-19 infection, comorbidities) in the vaccinated (one or more vaccines) population** (*N* = 192,604). Models built using boosted tree models (Catboost) with 5-fold cross-validation. Outcome is a binary event within follow-up y/n variable. 95% confidence intervals, derived from 1000x bootstrap resamples, are shown in shaded colour around the curves.

## Methods

The REACT-2 programme monitored the community prevalence of SARS-CoV-2 antibodies among adults in England, through a series of six large representative random samples of the population between July 2020 and May 2021. The study protocol[9] and summary reports for each round[1,2,19–21] have been published. Briefly: every 6–8 weeks, personalised invitations were sent to 315,000 adults aged 18 years and above on the NHS register, to achieve approximately equal numbers of responses from each of the 315 lower-tier local authority areas in England. Registration was done by telephone or online, and recruitment ceased when ~120,000 people had signed up. In the sixth and final round (May 2021), a boosted sample of adults aged 50 and above was obtained, with 749,225 invitations sent out resulting in 255,750 (34.1%) registrations and 209,482 (30.0%) completed tests.

Respondents were sent a LFIA (Fortress, United Kingdom) test kit for SARS-CoV-2 antibody self-testing and asked to report their test result and upload a photo of the completed test. Respondents also filled out a questionnaire including details of COVID-19 vaccination history, self-reported comorbidities and any history of suspected or confirmed COVID-19 (questionnaires available online at https://www.imperial.ac.uk/medicine/research-and-impact/groups/react-study/for-researchers/react-2-study-materials/).

Here, we use data on participants from rounds 5 (January–February 2021) and 6 (May 2021). We obtained Hospital Episode Statistics Admitted Patient Care data (until September 30, 2021) and Civil Registration Deaths data (until December 1, 2021) from NHS Digital for all participants with a valid LFIA result (positive or negative) who consented to this data linkage. We assessed the following outcomes from the date of completion of the LFIA to end of follow-up (December 1, 2021) or date of death: hospital admission for any reason (excluding admissions for pregnancy loss or childbirth, injuries, accidents or other external causes), unplanned/emergency admission for any reason (with the same exclusions), admission with COVID-19 as the primary diagnosis, admission where COVID-19 was listed anywhere on the record, all-cause mortality, deaths where COVID-19 was reported as the underlying condition, and deaths with COVID-19 mentioned anywhere on the death certificate (see Supplementary Methods S.1).

Participants' survey data were used to group people into age groups (50 years and under; 51–70 years; 71 years and over); male or female; previous history of COVID-19 (no, yes; confirmed, yes; probable or suspected); and self-reported vaccination status (0, 1, 2) with

people considered vaccinated 14 days after the reported date of vaccination. Participants who had received 3 or more vaccines at the time of the fieldwork were excluded from analysis. Participants were categorised as "high" or "low" risk based on the UK Joint Committee on Vaccination and Immunisation (JCVI) definition of being clinically extremely vulnerable for COVID-19[22,23] using self-reported comorbidity information (see Supplementary Methods S.2). Participants were also categorised into ethnic groups (Asian, Black, Mixed, Other, White); index of multiple deprivation (IMD) quintile using the residential postcode and current cigarette smoking status (yes, no). BMI was calculated from self-reported height and weight. Educational level was self-reported and grouped into categories.

### Statistical analyses

Using Kaplan–Meier plots we calculated the rate of each outcome by LFIA result and compared the cumulative hazard functions for those testing negative for the vaccinated group (*N* = 192,604) and for all participants, for each of the outcomes of interest. Plots were stratified by risk group, and then further stratified by age group and, for the full population, by vaccination status. We used multivariable Cox regression models to test the association between LFIA result and each outcome stratified by vaccination status in the full cohort. For each outcome, we constructed a series of nested models with increasing adjustments by age group, sex, (vaccination history), prior infection status and risk group. Hazard ratios (HRs) and 95% confidence intervals are reported.

To test whether the LFIA test result was adding substantially, as a predictor of health outcomes, to the information available in the survey (health, comorbidity and demographic), we used two multivariable approaches and included a range of potentially relevant covariates alongside the LFIA result. First, we used LASSO-penalised Cox regression models with stability analysis as implemented in the SHARP R package[24] to identify a parsimonious set of stably selected predictors for each of the outcomes. Selection thresholds were calibrated using an internal stability score. We conducted this analysis among individuals who had received one or more vaccinations and in the full cohort.

Second, we used extreme gradient boosted tree models as implemented in Catboost[25] to predict each of the binary outcomes in unseen data using a five-fold cross validation procedure. We evaluated various combinations of predictors (LFIA result only; covariates [age group, sex, risk group, self-reported vaccination status/count, self-

reported COVID-19 history] only; LFIA + covariates; LFIA + covariates + additional health and demographic data [BMI, smoking status, comorbidities, deprivation, education]) and evaluated model performance on unseen data using the area under the Receiver Operating Characteristic (ROC) curve (which provides the predictive accuracy of a binary classifier at different discrimination thresholds). Again, we conducted this analysis among individuals who had received one or more vaccinations and in the full cohort.

All data collection for the REACT2 study was captured with Questback. All data were analysed using R v4.0.5 (2021).

### Ethics
The REACT-2 study holds ethical approval from South Central−Berkshire B Research Ethics Committee (20/SC/0206; 21/SC/0163). Participants provided informed consent for their data to be used and, separately, indicated whether they were willing for their data to be linked to their NHS records.

### Reporting summary
Further information on research design is available in the Nature Portfolio Reporting Summary linked to this article.

### Data availability
The datasets generated or analysed, or both, during the current study are not publicly available because of governance restrictions and the identifiable nature of the data. Requests for access to raw data from the REACT study should be addressed to the corresponding authors and will be answered within 12 weeks. The linked administrative health data that support the findings of this study were provided by NHS Digital under license for the current study and cannot be made available by the study team. Data may be available by request through the NHS Digital Data Access Request Service (DARS) Process (https://digital.nhs.uk/services/data-access-request-service-dars/linked-datasets-supporting-health-and-care-delivery-and-research).

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

### Acknowledgements
We thank Anthony Thomas, Eric Johnson and Rob Elliott for their assistance in data acquisition, storage, preparation and governance. We acknowledge the REACT Public Advisory Panel who provided input into all stages of the research. All authors acknowledge Infrastructure support for the Department of Epidemiology and Biostatistics provided by the NIHR Imperial Biomedical Research Centre (BRC). Hospital Episode Statistics data are copyright © 2023, re-used with the permission of NHS Digital. All rights reserved. The Hospital Episode Statistics data were obtained from NHS Digital. The mortality data were supplied by NHS Digital derived by Office for National Statistics (ONS) from the national mortality and birth registrations and the Census. C.A.D. acknowledges support from the MRC Centre for Global Infectious Disease Analysis and

the NIHR Health Protection Research Unit in Emerging and Zoonotic Infections (NIHR HPRU award 200907). H.W. is a National Institute for Health Research (NIHR) Senior Investigator and acknowledges support from NIHR Biomedical Research Centre of Imperial College NHS Trust, NIHR School of Public Health Research, and NIHR Applied Research Collaborative North West London. G.C. is supported by a National Institute for Health Research (NIHR) Professorship. W.B. is the Action Medical Research Professor, A.D. is an NIHR senior investigator, and D.A. and P.E. are Emeritus NIHR Senior Investigators. P.E. is Director of the MRC Centre for Environment and Health (MR/L01341X/1, MR/S019669/1). P.E. acknowledges support from the NIHR Imperial Biomedical Research Centre and the NIHR HPRUs in Chemical and Radiation Threats and Hazards and in Environmental Exposures and Health, the British Heart Foundation Centre for Research Excellence at Imperial College London (RE/18/4/34215), Health Data Research UK (HDR UK) and the UK Dementia Research Institute at Imperial (MC_PC_17114). REACT-2 was funded by the Department of Health and Social Care in England. It is part of the Data and Connectivity National Core Study, led by Health Data Research UK in partnership with the Office for National Statistics and funded by UK Research and Innovation (grant ref MC_PC_20029).

## Author contributions

M.W.: conceptualisation, formal analysis, methodology, writing—original draft, data curation, visualisation; B.D.: conceptualisation, methodology, data curation, writing—original draft; C.A.: conceptualisation, methodology, writing—original draft; W.B.: conceptualisation, methodology, writing—original draft, supervision; D.A.: conceptualisation, project administration, supervision; A.D.: funding acquisition, project administration; S.R.: conceptualisation, funding acquisition; G.C.: conceptualisation, funding acquisition, writing—original draft; C.A.D.: conceptualisation, methodology, writing—original draft; M.C.-H.: conceptualisation, methodology, writing—original draft, supervision; P.E.: conceptualisation, methodology, writing—original draft, supervision, funding acquisition; H.W.: conceptualisation, methodology, writing—original draft, project administration, supervision. M.C.-H., P.E. and H.W. contributed equally.

## Competing interests

The authors declare no competing interests.
