## [Peer Review File · Nature Communications]

SARS-CoV-2 rapid antibody test results and subsequent risk of hospitalisation and death in 361,801 peopleREVIEWER COMMENTS

Reviewer #1 (Remarks to the Author):

Dear Sir,

Thank you for submitting this excellent manuscript with convincing evidence of the predictive value of self-administered LIFA tests for risk prediction of COVID-19 outcomes.

Major comments:

1) The data are nicely presented, but it would be fair for the reader to put more emphasis on the vaccinated group over the current presentation based on the entire cohort. Table 2 is misleading given that the important finding is the ones presented in Supplementary Table 3,4 and 5. It is not surprising to find a negative LIFA test in the non-vaccinated parts of the cohort. It would therefore be more accurate to focus on the results within the vaccinated group given that the clinical information from a negative test among vaccinated is much more specific as compared to a negative test among non-vaccinated individuals. You have a nice presentation of the effects/predictive value of LIFA negative among vaccinated individuals in supplementary table 3, 4, and 5 and supplementary figure 2. The supplementary table 5 shows that the negative LIFA associate with lower HR in non-vaccinated as opposed to the high HR in vaccinated individuals.

Minor changes:

1) Line 114: there is a "HR" missing in the bracket.

Reviewer #2 (Remarks to the Author):

The manuscript by Whitaker et al is a well executed study that assesses whether antibody rapid tests have predictive value for adverse health outcomes in a large general population population (n=361,801) of the UK. In particular, the manuscript investigates whether antibody rapid test results are associated with future all-cause and COVID-19 hospitalization and mortality. The authors shows that the negative results via the antibody rapid test was associated with elevated risk of hospitalization and death; and those who were "high-risk" and had a negative test also had elevated risk of all-cause mortality compared to high risk individuals who had positive tests. Unsurprisingly, antibody rapid tests results did not provide significant predictive information above other sociodemographic and health-related data.

Minor:

Discussion - 1) Can the investigators comment on why cumulative hazards was lower for several outcomes, particularly at the latest timepoints (Fig. 1)? 2) Authors should expand their discussion about the limitations with antibody rapid tests. The rapid test appears to have a sensitivity of ~67% for individuals with one or more vaccination at least 14 days prior to testing and goes up to ~85% in those that received two vaccinations. Many studies have demonstrated the low accuracy of antibody rapid tests, particularly in the immediate weeks following vaccination. In individuals with one or more vaccinations, those with a negative test had increased risk of hospitalization with COVID-19. Negative test results could be a result of low test sensitivity and/or low antibody titers. The authors should comment on these issues.

SARS-CoV-2 rapid antibody test results and subsequent risk of hospitalisation and death in 361,801 people: response to reviewers

Reviewer 1

Major comments:

1) The data are nicely presented, but it would be fair for the reader to put more emphasis on the vaccinated group over the current presentation based on the entire cohort. Table 2 is misleading given that the important finding is the ones presented in Supplementary Table 3,4 and 5. It is not surprising to find a negative LIFA test in the non-vaccinated parts of the cohort. It would therefore be more accurate to focus on the results within the vaccinated group given that the clinical information from a negative test among vaccinated is much more specific as compared to a negative test among non-vaccinated individuals. You have a nice presentation of the effects/predictive value of LIFA negative among vaccinated individuals in supplementary table 3, 4, and 5 and supplementary figure 2. The supplementary table 5 shows that the negative LIFA associate with lower HR in non-vaccinated as opposed to the high HR in vaccinated individuals.

Thank you for this comment. We agree that the results in the vaccinated cohort are more relevant. We have therefore foregrounded the results in the vaccinated population. Specifically, we have:

- Replaced the K-M plots in Figure 1 with plots showing the vaccinated (1+) population only.
- Swapped main figure 2 and supplementary figure 2 (so that the stability selection results presented in the main are for a vaccinated population)
- Swapped main figure 3 and supplementary figure 3 (so that the boosted tree prediction results presented in the main are for a vaccinated population)
- Amended the results to focus on HR results in the vaccinated (1+) population
- Amended the abstract to include HRs and risk/1000 numbers from the vaccinated (1+) population
- Added a paragraph comparing all-cause mortality risk/1000 in LFIA negatives vs positives in the vaccinated population

Minor changes:

1) Line 114: there is a "HR" missing in the bracket.

Thank you. Corrected.

Reviewer 2

Minor:

Discussion - 1) Can the investigators comment on why cumulative hazards was lower for several outcomes, particularly at the latest timepoints (Fig. 1)?

In line with recommendations from reviewer 1, we have now replaced Fig 1 with an equivalent plot showing the vaccinated population only. In this plot the cumulative hazards diverge in a more intuitive manner. The slight convergence that was evident in the full-cohort K-M plots was likely to be attributable to bias arising from the relationships between risk level/age, vaccination status, and likelihood of testing LFIA positive.

2) Authors should expand their discussion about the limitations with antibody rapid tests. The rapid test appears to have a sensitivity of ~67% for individuals with one one or more vaccination at least 14 days prior to testing and goes up to ~85% in those that received two vaccinations. Many studies have demonstrated the low accuracy of antibody rapid tests, particularly in the immediate weeks following vaccination. In individuals with one or more vaccinations, those with a negative test had increased risk of hospitalization with COVID-19. Negative test results could be a result of low test sensitivity and/or low antibody titers. The authors should comment on these issues.

Thank you for raising these points. We have expanded the limitations section to address them.